# Association of Single Nucleotide Polymorphisms Located in *LOXL1* with Exfoliation Glaucoma in Southwestern Sweden

**DOI:** 10.3390/genes12091384

**Published:** 2021-09-03

**Authors:** Marcelo Ayala, Madeleine Zetterberg, Ingmar Skoog, Anna Zettergren

**Affiliations:** 1Department of Clinical Neuroscience, Institute of Neuroscience and Physiology, Sahlgrenska Academy, University of Gothenburg, 40530 Gothenburg, Sweden; madeleine.zetterberg@vgregion.se; 2Eye Department, Region Västra Götaland, Skaraborg Hospital/Skövde, 54142 Skövde, Sweden; 3Department of Clinical Neuroscience, Karolinska Institute, 17165 Stockholm, Sweden; 4Department of Ophthalmology, Region Västra Götaland, Sahlgrenska University Hospital, 43130 Mölndal, Sweden; 5Region Västra Götaland, Sahlgrenska University Hospital, Psychiatry, Cognition and Old Age Psychiatry Clinic, 40530 Gothenburg, Sweden; ingmar.skoog@neuro.gu.se; 6Neuropsychiatric Epidemiology Unit, Department of Psychiatry and Neurochemistry, Institute of Neuroscience and Physiology, The Sahlgrenska Academy, Centre for Ageing and Health (AGECAP) University of Gothenburg, 40530 Gothenburg, Sweden; anna.zettergren@neuro.gu.se

**Keywords:** allelic frequency, *LOXL1*, exfoliation glaucoma, Sweden

## Abstract

Introduction: Glaucoma is an optic neuropathy that leads to visual field defects. Genetic mechanisms seem to be involved in glaucoma development. Lysyl Oxidase Like 1 (*LOXL1*) has been described in previous studies as a predictor factor for exfoliation glaucoma. The present article studied the association between three single nucleotide polymorphisms (SNPs) in the *LOXL1* gene and the presence of exfoliation glaucoma in Southwestern Sweden. Methods: Case-control study for genetic association. In total, 136 patients and 1011 controls were included in the study. Patients with exfoliation glaucoma were recruited at the Eye Department of Sahlgrenska University Hospital and Skaraborgs Hospital, Sweden. Controls were recruited from the Gothenburg H70 Birth Cohort Study. Three different SNPs were genotyped: *LOXL1*_rs3825942, *LOXL1*_rs2165241 and *LOXL1*_rs1048661. Results: The distribution of allele frequencies was significantly different between controls and glaucoma patients; for rs3825942 (*p* = 2 × 10^−12^), for rs2165241 (*p* = 3 × 10^−16^) and for rs1048661 (*p* = 2 × 10^−6^). Logistic regression analyses using an additive genetic model, adjusted for sex and age, also showed associations between the studied SNPs and glaucoma (*p* = 9 × 10^−6^; *p* = 2 × 10^−14^; *p* = 1 × 10^−4^). Conclusion: A strong association was found between allele frequencies of three different SNPs (*LOXL1*_rs3825942, *LOXL1*_rs2165241, and *LOXL1*_rs1048661) and the presence of exfoliation glaucoma in a Southwestern Swedish population.

## 1. Introduction

Glaucoma is an optic neuropathy that leads to impaired vision, first in the peripheral visual field, but it may also decrease central visual acuity as the disease progresses. In western countries, glaucoma is the most common cause of blindness [1]. The disease is chronic and is characterized by a loss of ganglion cells with subsequent visual field loss. Even today, no treatment has been discovered to cure glaucoma. The cause of glaucoma still remains unknown, but several risk factors have been identified, of which increased intraocular pressure (IOP) is the most common [2]. There are different types of glaucoma; the two most common in Scandinavia are primary open-angle (POAG) and exfoliation glaucoma (EXFG) [3].

Exfoliation glaucoma (EXFG) is characterized by the presence of exfoliative material in the anterior chamber of the eye. Exfoliations are composed of proteins, but the origin of the material is unknown. Exfoliative material has also been isolated in other parts of the eye and even in extraocular tissues. In the eye, the pupillary margin, the anterior part of the iris, the trabecular meshwork, and the anterior capsule of the lens are the most common sites where exfoliation material is displayed. Exfoliations at the trabecular meshwork may occlude the pores diminishing the outflow of aqueous humor, thus increasing the IOP [4,5,6,7]. Exfoliation material can be found in many different organs. Systemic associations include ischemic attacks, hypertension, angina, myocardial infarction, cerebrovascular and cardiovascular disease, aortic aneurysm, Alzheimer’s disease, and hearing loss [6]. 

As described above, the etiology of exfoliations is still unknown, but several studies suggest an association with genetic factors [8,9,10,11]. The most frequently studied gene related to exfoliation syndrome is Lysyl Oxidase Like 1 (*LOXL1*). The *LOXL1* gene encodes an enzyme (Lysyl Oxidase) essential to the biogenesis of connective tissue by catalyzing the first step in collagen and elastin formation. Thorleifsson et al. published a genome-wide association study (GWAS) in 2007 that was the first genetic association study on glaucoma including Swedish individuals (in combination with Icelandic individuals) [12]. The authors pointed out three different SNPs as related to exfoliation glaucoma. 

The present study aimed at investigating the association between three single nucleotide polymorphisms (SNPs) in the *LOXL1* gene and the presence of exfoliation glaucoma in patients living in Southwestern Sweden. 

## 2. Materials and Methods

The present study was a case-control study to investigate genetic associations between SNPs in the *LOXL1* gene and a diagnosis of EXFG. The study adhered to the tenets of the declaration of Helsinki. All patients signed informed consent. The study was approved by the Regional Ethics Review Board in Gothenburg. 

### 2.1. Subjects

Exfoliation glaucoma patients were prospectively recruited at the Eye Department, Skaraborg Hospital, Skövde, and at the Department of Ophthalmology, Sahlgrenska University Hospital, Mölndal, Sweden. Both hospitals are in the southwestern part of Sweden; Region Västra Götaland. In addition, control individuals were recruited from the general population within the Gothenburg H70 Birth Cohort Study, in which individuals being 70 years and living in Gothenburg were invited to participate. The H70 study has been described in detail elsewhere [13,14].

In total, 1203 controls were available for this study. Among them, 149 individuals were excluded because they were not ethnically Swedish or Finnish, nor their parents. Further, forty-three individuals on glaucoma medications and/or with a previous glaucoma diagnosis were excluded. Thus, the final number of included controls was 1011 individuals. Among the glaucoma cases, three subjects and/or their parents were born in Finland, and the rest (*n* = 133) in Sweden. Thus, in total, 136 exfoliation glaucoma patients were included in the study. 

### 2.2. Ophthalmic Examinations

For glaucoma cases, a detailed medical and ocular history was recorded for each patient. Visual acuity, intraocular pressure (IOP), gonioscopy, optic nerve status, visual field examinations, current medications, and the presence or absence of exfoliations were recorded. A Snellen chart was used to measure visual acuity. The IOP was measured by Goldmann tonometry. To assess trabecular meshwork, gonioscopy was performed in a dark room using a goniolens with an undilated pupil. The presence of Sampaolesi’s line was also recorded at the inferior angle. For fundus examination, 2.5% Phenylephrine and Tropicamide 0.5% (Bausch and Lomb U.K. Ltd., London, UK) were used. Exfoliations, if present, were detected on the pupil, anterior lens surface, or the anterior chamber angle with dilated pupils. The optic nerve appearance was studied using a 90-D lens, and the average vertical cupping was recorded as the cup-to-disc ratio. All patients were subjected to repeated Humphrey Field Analysis (Carl Zeiss, Carl-Zeiss-Straße 22, 73447 Oberkochen, Germany) using the software threshold 24-2. Only reliable visual fields were considered (see above). Topical IOP-lowering treatment was denoted as the number of pharmacological substances. If both eyes were suffering glaucoma, one eye was chosen at random.

Exfoliation glaucoma was defined as untreated IOP of 21 mm Hg or higher, an open anterior chamber angle at gonioscopy, glaucomatous visual field defect (at least two repeatable examinations with Humphrey 24-2), and glaucomatous optic nerve damage, concomitant with the presence of exfoliation material, observed at the anterior lens capsule and/or at the pupillary border according to the definition of the European Glaucoma Society [15]. 

Controls were asked if they were on IOP-lowering medications or if they were previously diagnosed with glaucoma. In the case of a positive answer, the individuals were excluded from statistical analyses. No ophthalmological examination was performed among controls. 

### 2.3. Genetic Analysis

All patients underwent venipuncture for blood sample extraction. Extraction of DNA from blood samples was performed according to standard procedures. Genotyping of three SNPs (rs3825942, rs2165241, and rs1048661) in the *LOXL1* gene was performed at LGC Genomics (Hoddesdon, Herts, UK), using the KASPar PCR SNP genotyping system. The success rate was over 95% for all genotyped SNPs, and they were in Hardy–Weinberg equilibrium.

### 2.4. Statistics

For statistical analysis, SPSS (IBM, 1 New Orchard Road Armonk, NY 10504, USA) software was used. Baseline age was tested for normality using the Kolmogorov–Smirnov test and homoscedasticity using Levene’s test. If data were not normally distributed, Mann–Whitney U-test was used (controls vs. glaucoma patients). The distribution of the categorical variable gender was tested using the chi-square test. 

The SNP distribution between cases (glaucoma patients) and controls was tested using the chi-square test and Fischer’s exact test. In addition, logistic regression was performed, using both an unadjusted model and a model including age and sex as covariates. The genotype data were coded as follows: non-risk homozygote = 0, heterozygote = 1, risk homozygote = 2 for the additive genetic model; non-risk homozygote = 0, heterozygote + risk homozygote = 1 for the dominant genetic model; and non-risk homozygote + heterozygote = 0, risk homozygote = 1 for the recessive genetic model. A Bonferroni corrected *p*-value was calculated multiplying the obtained *p*-value by three (due to three different SNPs tested). 

A power analysis based on Fisher’s test was performed using the Power and Sample Size Calculations (P.S.) software. (http://biostat.mc.vanderbilt.edu/PowerSampleSize, accessed on 25 August 2021). Post-hoc power analysis using 136 cases and 1011 control subjects was carried out. Each SNP was analysed separately. In all cases the Type I error was set to 0.05. 

## 3. Results

Age at baseline showed no normality according to the Kolmogorov–Smirnov test (*p* < 0.0001). In addition, Levene’s test showed no homoscedasticity (*p* = 0.003). A Mann–Whitney U-test demonstrated a significant difference in age when comparing the control individuals vs. glaucoma patients (*p* < 0.001). Gender distribution did not differ between glaucoma and control groups (Table 1). 

The three studied SNPs genotype distribution was found to be significantly different between controls and exfoliation glaucoma patients. For rs3825942, the G:G genotype was present among 99.2% of the individuals with EXFG and 72.4% of the controls. For rs2165241, the T:C showed the highest frequency among controls (50.4%), while the T:T showed the highest frequency among the EXFG (56%). Finally, for rs1048661, the T:G showed the highest frequency among controls (44.3%), while the G:G showed the highest frequency among the EXFG (61.8%) (Table 2).

Significant differences were found regarding the minor allele frequencies when comparing the controls and the EXFG. The minor allele frequencies were higher in controls than in EXFG patients in rs3825492 and rs1048661 but not in rs2165241. See Table 3. 

Post-hoc power analyses showed that the power for rs3825942 was 0.99, for rs2165241 0.99 and, for rs1048661 0.87.

The relation between the SNPs and glaucoma diagnosis was also analyzed using binary logistic regression analysis. In the first model (unadjusted), glaucoma (yes/no) was the dependent variable, and genotype was the independent variable. In the second model (adjusted), glaucoma was the dependent variable, and genotype, age, and sex were independent variables. The three SNPs were investigated in three separate genetic models: additive, dominant and recessive. Based on the additive genetic model, all SNPs showed significant values (Table 4). The dominant genetic model showed only significant values for rs2165241 (Table 5). The recessive genetic model showed significant results for rs3825942 and rs2165241 but not for rs1048661 (Table 6). 

## 4. Discussion

The present study demonstrates associations between the SNPs *LOXL1*_rs3825942, *LOXL1*_rs2165241, *LOXL1*_rs1048661, and a diagnosis of exfoliation glaucoma. The only previous study including individuals living in Sweden was published by Thorleifsson et al. in 2007 [12]. The present study shows similar results to the reports by Thorleifsson et al., although the ORs coming from the present study were slightly lower. Thorleifsson’s study included controls from Iceland predominantly, and cases were glaucoma patients from the Uppsala region, Sweden, and Iceland. Glaucoma patients were included retrospectively based on medical records, and controls were provided through anonymous blood samples. In the present study, cases were included prospectively, diminishing the possibility of bias. In addition, inclusion was based on a strict protocol, and an ophthalmologist controlled all cases to certify the EXFG diagnosis before inclusion. Other differences between the studies were that the present study recruited both controls and cases from the same region, while in the study by Thorleifsson et al., the majority of controls were from Iceland. In contrast, the majority of glaucoma patients were from Sweden. 

Exfoliation syndrome and exfoliation glaucoma have been described worldwide. However, the prevalence of the phenotype varies among countries. For example, the prevalence is high in Scandinavia but low in China. Åström et al. reported a prevalence of 23% among Swedes at 63 years of age, while the prevalence was 0.55% in a Chinese population of 60 years or older [16,17]. Studies have shown that the influence of the three studied SNPs on glaucoma risk differs depending on ethnicity. For the rs2165241, the T allele was a risk factor in studies based on European populations [12,18,19,20], while in Asian studies, it was the opposite (i.e., the high frequency of allele C was the risk factor for the development of exfoliation) [21]. Additionally, for rs3825942, the high frequency of the G allele was the risk factor for developing exfoliation in the present study and previous studies based on European populations [12,18,19,20]. However, the opposite was described in a study performed in South Africa, including black subjects, where the authors reported the A allele as a risk factor for exfoliation syndrome [22]. The present study took into consideration the ethnicity of the participants. All participants, both among the cases and the controls, were questioned about their and their parents’ birthplace. Most exfoliation glaucoma patients were born in Sweden, and a minority (around 3%) were born in Finland. The control group included a similar distribution of individuals and excluded patients born in other countries. A Finland-based study showed a lower OR for the G allele of rs3825942 than the present study, but still with an association in the same direction. Similar results were described for the two other SNPs [18]. The small number of Finnish patients and controls included in the present study will probably not significantly influence the results. 

The frequency of the studied alleles in the control group should be discussed. For the three different SNPs studied, risk alleles showed a high frequency even among control individuals. For rs3825942, the G allele frequency (risk allele) was 82.5% among controls. For rs2165241, the C allele frequency (risk allele) was 51% in controls, and for rs1048661, the G allele frequency (risk allele) was 66.3% in controls. Based on the relatively high frequency of the risk alleles among healthy individuals found in this study, it is apparent that the risk alleles themselves cannot fully explain the disease’s presence. As with all complex genetic disorders, other factors apart from common SNPs can probably be implicated in the development of the disease. For example, environmental factors might be considered. Furthermore, the present study investigated only one gene (*LOXL1*); several other genes have been described as associated with the disease [23]. 

The location of the *LOXL-1* gene and its function have been widely described in previous studies [24]. The gene localizes in chromosome 15 (15q24.1) and encodes a member of the lysyl oxidase family of proteins (https://www.ncbi.nlm.nih.gov/gene/4016 accessed on 25 August 2021). The lysyl oxidase enzyme is involved in the biogenesis of connective tissue by crosslinking the extracellular matrix proteins, collagen, and elastin [25]. Previous studies have shown a possible down-regulation of *LOXL-1* as a mechanism for exfoliation formation in EXFG [26]. Decreased activity of the LOXL-1 enzyme increases elastin levels which is an essential part of the exfoliation material [27]. The detailed pathways from the gene and its SNPs to exfoliation material are still not elucidated. Probably, the SNPs act by altering the DNA expression of the *LOXL-1,* inducing an increased elastin level in the anterior chamber of the eye. 

The three different SNPs evaluated in the present study are localized in different parts of the *LOXL-1* gene. The rs3825942 and rs1048661 are nonsynonymous SNPs located in exon 1 [28], while rs2165241 is a non-coding SNP located in intron 1 [20]. The three SNPs were chosen based on previous studies [21,29,30]. According to Thorleifsson et al. [12], the linkage between these three SNPs is low to medium (r^2^ values ranging from 0.13–0.45 with possible values for r2 within the range 0–1). It is apparent that the three SNPs contribute with independent information, and it is valuable to include all of them in a study like the present one. 

The present study has some limitations. One limitation is how suspected glaucoma subjects were excluded from the control group. The exclusion was based upon self-reported disease. Therefore, it is possible that some of the controls had yet undiagnosed glaucoma and were incorrectly included as controls. However, the large number of controls included may alleviate a possible effect of the procedure used for excluding glaucoma cases from the control group. 

Another limitation of the study is that all included controls and patients were ethnically Swedish or Finnish. Results coming from the present study may therefore not be applied to other populations. In addition, the number of cases and controls included was relatively small, considering the sample size often used in GWAS. However, the phenotype was very well defined, and the effect sizes of the identified associations were similar to those reported in a previous GWAS [12]. Another limitation is that additional genes/SNPs of potential importance for glaucoma were not considered in this study.

## 5. Conclusions

In conclusion, the present study, using modern genetic techniques and a defined phenotype, confirms previous studies’ results. The study showed a strong association between the SNPs *LOXL1*_rs3825942, *LOXL1*_rs2165241 *LOXL1*_rs1048661, and the development of exfoliation glaucoma in a Swedish population located in Southwestern Sweden. 

## Figures and Tables

**Table 1 genes-12-01384-t001:** Baseline demographic of the individuals included in the study.

	Control (*n* = 1011)	EXFG (*n* = 136)	Test	*p*-Values
Age (mean) (SD)	70.55 (0.28)	73.02 (6.16)	Mann–Whitney	<0.001 *
Sex (M/F) (%)	457/554 (45/55)	62/74 (46/54)	chi-square	0.46

EXFG: exfoliation glaucoma. SD: standard deviation. M: male. F: female. * A *p*-value < 0.05 was considered statistically significant.

**Table 2 genes-12-01384-t002:** Genotype frequency distribution between controls and exfoliation glaucoma patients.

Rs-ID	Genotype	Control (*n* = 1011)	EXFG (*n* = 136)	*p*-Values
rs3825942	G:G (*n*) (%)	732 (72.4)	135 (99.2)	7 × 10^−11^ *
	G:A (*n*) (%)	256 (25.3)	1 (0.7)	
	A:A (*n*) (%)	23 (2.3)	0 (0)	
rs2165241	C:C (*n*) (%)	260 (25.7)	9 (6.7)	3 × 10^−15^ *
	T:C (*n*) (%)	510 (50.4)	51 (37.3)	
	T:T (*n*) (%)	241 (23.9)	76 (56)	
rs1048661	G:G (*n*) (%)	445 (44)	84 (61.8)	3 × 10^−4^ *
	T:G (*n*) (%)	448 (44.3)	45 (33.1)	
	T:T (*n*) (%)	118 (11.6)	7 (5.1)	

EXFG: exfoliation glaucoma, * A *p*-value < 0.05 was considered statistically significant. The chi-square test was used to test significance.

**Table 3 genes-12-01384-t003:** Minor allele frequencies of SNPs in *LOXL1*.

Rs-ID	Minor Allele ^(1)^	Control (*n* = 2022)	EXFG ^(2)^ (*n* = 272)	*p*-Values
rs3825942	A < G (*n*) (%)	354 17.5	2 0.7	2 × 10^−12^ *
rs2165241	T < C (*n*) (%)	989 48.9	203 74.6	3 × 10^−16^ *
rs1048661	T < G (*n*) (%)	679 33.7	59 21.7	2 × 10^−6^ *

^(1)^ The minor allele was determined based on the allele frequency among controls. ^(2)^ EXFG: exfoliation glaucoma.* A *p*-value < 0.05 was considered statistically significant. The Fisher-exact test was used to test significance.

**Table 4 genes-12-01384-t004:** Associations between the SNPs and glaucoma based on logistic regression analysis, additive genetic model.

Rs-ID	Model Unadjusted for Age and Sex		Model Adjusted for Age and Sex	
	Odds Ratio (Exp β) (95% Conf. Int)	*p*-Values (Corrected *p*-Values ^1^)	Odds Ratio (Exp β) (95% Conf. Int)	*p*-Values (Corrected *p*-Values ^1^)
rs3825942	23.61 (5.83–95.56)	5 × 10^−5^ * (1 × 10^−4^) *	17.91 (4.41–72.69)	9 × 10^−^^6^ * (2 × 10^−5^) *
rs2165241	3.16 (2.35–4.25)	2 × 10^−11^ * (6 × 10^−11^) *	2.96 (2.16–4.07)	2 × 10^−14^ * (6 × 10^−14^) *
rs1048661	1.82 (1.31–2.51)	3 × 10^−4^ * (9 × 10^−4^) *	1.82 (1.34–2.46)	1 × 10^−^^4^ * (3 × 10^−4^) *

* A *p*-value < 0.05 was considered statistically significant. ^1^ Bonferroni-corrected *p*-values for multiple testing.

**Table 5 genes-12-01384-t005:** Associations between the SNPs and glaucoma based on logistic regression analysis, dominant genetic model.

Rs-ID	Model Unadjusted for Age and Sex		Model Adjusted for Age and Sex	
	Odds Ratio (Exp β) (95% Conf. Int)	*p*-Values (Corrected *p*-Values ^1^)	Odds Ratio (Exp β) (95% Conf. Int)	*p*-Values (Corrected *p*-Values ^1^)
rs3825942	4.53 × 10^−9^	0.99 (N.A.)	5.67 × 10^−9^	0.99 (N.A.)
rs2165241	4.84 (2.09–11.21)	3 × 10^−4^ * (9 × 10^−4^) *	4.11 (1.74–9.71)	1 × 10^−3^ * (3 × 10^−3^) *
rs1048661	2.88 (1.04–7.99)	5 × 10^−2^ * (0.15)	2.51 (0.86–7.29)	0.09 (0.27)

* A *p*-value < 0.05 was considered statistically significant. ^1^ Bonferroni-corrected *p*-values for multiple testing.

**Table 6 genes-12-01384-t006:** Associations between the SNPs and glaucoma based on logistic regression analysis, recessive genetic model.

Rs-ID	Model Unadjusted for Age and Sex		Model Adjusted for Age and Sex	
	Odds Ratio (Exp β) (95% Conf. Int)	*p*-Values (Corrected *p*-Values ^1^)	Odds Ratio (Exp β) (95% Conf. Int)	*p*-Values (Corrected *p*-Values ^1^)
rs3825942	25.53 (6.27–103.86)	6 × 10^−6^ * (2 × 10^−5^) *	19.28 (4.72–78.75)	4 × 10^−5^ * (1 × 10^−4^) *
rs2165241	0.20 (0.09–0.48)	2 × 10^−4^ * (6 × 10^−4^) *	0.24 (0.11–0.57)	1 × 10^−3^ * (3 × 10^−3^) *
rs1048661	0.35 (0.12–0.96)	4 × 10^−2^ * (0.12)	0.40 (0.18–1.15)	0.09 (0.27)

* A *p*-value < 0.05 was considered statistically significant. ^1^ Bonferroni-corrected *p*-values for multiple testing.

## Data Availability

A summary of anonymous data can be required from the correspondent author.

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
