# Peer review of "Association of Single Nucleotide Polymorphisms Located in LOXL1 with Exfoliation Glaucoma in Southwestern Sweden"

_genes, 2021, doi:10.3390/genes12091384_

Round 1

Reviewer 1 Report

The present article studied the association between three single nucleotide polymorphisms (SNPs) in the LOXL1 gene and the presence of exfoliation glaucoma (EXFG) in Southwestern Sweden. This study showed a strong association between the SNPs LOXL1_rs3825942, LOXL1_rs2165241 LOXL1_rs1048661, and the development of exfoliation glaucoma in a Swedish population located in Southwestern Sweden. The topic of this study seems to be clinically relevant. However, there are several issues the authors need to address.

1. A power analysis is required to demonstrate that this study has sufficient power to detect the associations of these SNPs with EXFG.

2. It is not clear how the three SNPs in the LOXL1 gene were selected. What criteria are used to select these three SNPs? Do they cover the whole LD of the LOXL1 gene?

3. The authors are advised to consider many possible genetic models, including additive, dominant, recessive, and co-dominant. 

4. Multiple testings should be appropriately corrected.

Author Response

  1. A power analysis is required to demonstrate that this study has sufficient power to detect the associations of these SNPs with EXFG.

Answer: I agree. A power calculation was added. Please see both in Methods and Results section.

  1. It is not clear how the three SNPs in the LOXL1 gene were selected. What criteria are used to select these three SNPs? Do they cover the whole LD of the LOXL1 gene?

Answer: I agree. The study was based on previous published data. It’s possible to assume that these three SNPs covered the whole LD of the LOXL1 genes. Two new sentences were added in the introduction section.

  1. The authors are advised to consider many possible genetic models, including additive, dominant, recessive, and co-dominant. 

Answer: I agree, we added a dominant and a recessive genetic model. The co-dominant model was judged not to add further information to the study. Please see table 5 and 6.

  1. Multiple testings should be appropriately corrected.

Answer: I agree. A Bonferroni-corrected p-value has been added.

Reviewer 2 Report

The manuscript is well written and illustrated. I would like to congratulate the Authors for their excellent work. 

thank you for associating me with the revision of this paper  studing the  association between allele frequencies of three different SNPs in LOXL1 gene and the presence of exfoliation glaucoma in a Southwestern Swedish population. I have no comments to add except that his is an interesting and very well written article that targets the swedish population and could in this case be difficult to apply to other populations. Also, one of the limitations which should be specified is that no ophthalmological examination was performed among controls. Autors should precise criteria for the choice of the 3 studied SNPs.  Finally, in the exfoliative glaucoma group, is it possible to identify the differences between the subgroups according to the selected SNPs.

Author Response

  • Also, one of the limitations which should be specified is that no ophthalmological examination was performed among controls.

Answer: I agree. This has now been clarified.

  • Authors should precise criteria for the choice of the 3 studied SNPs. 

Answer: I agree. Two new sentences were added in the introduction section.

  • Finally, in the exfoliative glaucoma group, is it possible to identify the differences between the subgroups according to the selected SNPs